# In-situ monitoring of interface proximity effects in ultrathin ferroelectrics

Nives Strkalj [1,4 ✉], Chiara Gattinoni[1], Alexander Vogel [2,4], Marco Campanini [2], Rea Haerdi[1], Antonella Rossi[1,3], Marta D. Rossell [2], Nicola A. Spaldin [1], Manfred Fiebig [1] & Morgan Trassin [1 ✉]

The development of energy-efficient nanoelectronics based on ferroelectrics is hampered by a notorious polarization loss in the ultrathin regime caused by the unscreened polar discontinuity at the interfaces. So far, engineering charge screening at either the bottom or the top interface has been used to optimize the polarization state. Yet, it is expected that the combined effect of both interfaces determines the final polarization state; in fact the more so the thinner a film is. The competition and cooperation between interfaces have, however, remained unexplored so far. Taking $PbTiO_3$ as a model system, we observe drastic differences between the influence of a single interface and the competition and cooperation of two interfaces. We investigate the impact of these configurations on the $PbTiO_3$ polarization when the interfaces are in close proximity, during thin-film synthesis in the ultrathin limit. By tailoring the interface chemistry towards a cooperative configuration, we stabilize a robust polarization state with giant polarization enhancement. Interface cooperation hence constitutes a powerful route for engineering the polarization in thin-film ferroelectrics towards improved integrability for oxide electronics in reduced dimension.

[1] Department of Materials, ETH Zurich, 8093 Zurich, Switzerland. [2] Electron Microscopy Center, Swiss Federal Laboratories for Materials Science and Technology, Empa, 8600 Dübendorf, Switzerland. [3] Department of Chemical and Geological Sciences, University of Cagliari, 09124 Cagliari, Italy. [4] These authors contributed equally: Nives Strkalj, Alexander Vogel. ✉email: nives.strkalj@mat.ethz.ch; morgan.trassin@mat.ethz.ch

The two interfaces of a thin film are instrumental in setting its properties. Tight control over the interface configuration is therefore vital for applications. For example, in ferroelectric materials, a major obstacle to stabilizing the electric polarization in ultrathin layers is uncompensated bound charge at the interfaces[1–4]. Approaches to combat the resulting depolarizing field focus on acting on the bound charge at either the bottom interface by introducing metallic buffer electrodes[2,4] and atomic-scale interface engineering[5], or at the top interface by introducing charge-screening environments such as gases[6,7] or liquids[8–10]. Acting on both interfaces in the same experiment is hampered by the lack of separate, yet simultaneous experimental access for observing how these interfaces set the resulting polarization state of the film. Understanding such correlation becomes especially significant when the interfaces are separated by only a few nanometers.

Combined interface effects in ferroelectric thin films are most directly accessed by observing the spontaneous polarization right when it is formed—during assembly of the film in the growth chamber[11,12]. Using a laser-optical detection technique, in situ second harmonic generation (ISHG)[13], we achieve this in ferroelectric PbTiO$_3$ (PTO). We use PTO as a model system to explore the joint effects of two nanometer-spaced interfaces. We design and identify the influence of the bottom layer for itself and compare it to the influence exerted jointly by the PTO bottom and top interfaces, which we then relate to the resulting net polarization. Microscopic understanding is gained through scanning transmission electron microscopy (STEM), angle-resolved X-ray photoelectron spectroscopy (XPS), and first-principles calculations. Competitive and cooperative interface configurations are achieved by engineering the termination of the bottom interface to cause the film to exhibit either upwards or downwards polarization, while the top interface always promotes downwards polarization. In the cooperative case, giant polarization enhancement occurs when the influence of the top interface complements that of the bottom layer. Such fundamental understanding of proximity effects of thin-film interfaces should help to promote the integration of ultrathin ferroelectrics beyond previous limitations.

## Results

We begin our investigation by monitoring the emergence of polarization in a PTO film using ISHG[13]. SHG denotes frequency doubling of a light wave in a material, a process sensitive to the absence of inversion symmetry. It therefore occurs with the onset of ferroelectric order, and with an amplitude that is proportional to the dipole moment and thickness in thin films[14,15]. All ISHG measurements were conducted under identical experimental conditions, see "Methods," and the resulting intensities in arbitrary units can thus be quantitatively compared. Simultaneously, reflection high energy electron diffraction (RHEED) is performed to ensure the growth quality and to calibrate the ISHG yield to the film thickness with unit-cell (u. c.) accuracy.

The PTO film grown on SrTiO$_3$ (STO) substrate is buffered by a La$_{0.7}$Sr$_{0.3}$O-terminated (001)-oriented metallic La$_{0.7}$Sr$_{0.3}$MnO$_3$ (LSMO) layer which sets the macroscopic single-domain polarization upwards, away from the bottom interface[5]. Details of the growth of the ferroelectric films are reported in Supplementary Note 1. During PTO deposition, we observe a rather striking ISHG evolution (Fig. 1a). (i) Polarization emerges from the very first u. c., with subsequent increase of ISHG intensity with thickness. (ii) Upon interrupting the deposition, the ISHG evolution abruptly changes and the ISHG intensity gradually decreases to a new stable value, see Fig. 1b. (iii) Upon resuming the deposition, the ISHG intensity abruptly recovers to its pre-interruption value and continues to increase with thickness as in (i).

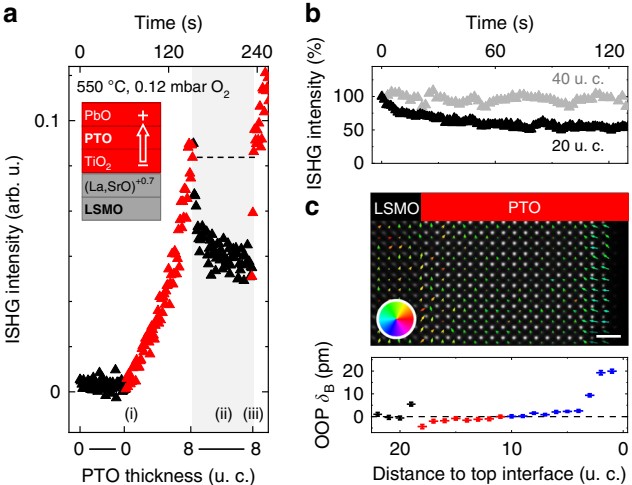

**Fig. 1 Polarization suppression from competitive interfaces. a** ISHG signal tracking the PTO thin-film polarization during ongoing (filled red symbols, 0–8 u. c.) and halted (filled black symbols) growth. The time axis for this growth protocol reveals relaxation of the polarization on the order of 10$^2$ s. The inset shows the chemistries of interfacial planes at the PTO|LSMO interface and the polarization direction set by the bottom interface. The dashed line indicates recovery of the ISHG intensity upon resuming the deposition. **b** Time-dependent ISHG signal after interrupting deposition, normalized to the maximum value = 100. **c** Post-deposition STEM map of the dipole moments. The arrows show the direction (color wheel) and amplitude (arrow length) of the associated dipole moments (top). Scale bar is 1 nm. The OOP B-site displacement ($\delta_B$) of the persistent state is mapped throughout the PTO|LSMO bilayer (bottom). Black symbols refer to LSMO, red and blue symbols refer to upwards and downwards polarization, respectively, in PTO. The error bars are the standard error of the mean. The electrostatic boundary conditions at the La$_{0.7}$Sr$_{0.3}$O|TiO$_2$-terminated LSMO|PTO interface result in the observed displacement discontinuity, see main text and ref. [16].

The observed absence of a critical thickness (i) has been previously reported for Pb-based ferroelectrics[16–19], and is further highlighted in Supplementary Note 2. The ISHG drop at the growth interruption (ii) is only observed for PTO films thinner than 40 u. c. (see Fig. 1b). The origin of the ISHG drop could be sample degradation, polarization suppression, or domain formation[4,20]. We therefore conduct a post-deposition STEM analysis by mapping the atomic displacements throughout the PTO film. The local manifestation of the polarization drop (ii) in the high-quality films is twofold, see Fig. 1c. First, dipole moments next to the top interface are reversed with respect to the direction set by the bottom interface, pointing downwards across 3–4 u. c. Second, beneath this region, the dipole moments retain the orientation set by the bottom interface, but their magnitude is suppressed, with PTO out-of-plane (OOP) B-site displacements of <5 pm compared to 16 pm in the bulk[21]. This points to the top interface favoring downwards polarization and therefore to a competition between bottom and top interfaces in setting the net polarization direction. This strikingly results in a substantial suppression of the polarization throughout the film.

We observe in Fig. 1b that the process responsible for the influence of the top interface on the polarization state occurs on a much longer time scale with respect to the growth process. As long as the growth continues, the polarization state is therefore essentially determined by a single interface, the bottom interface. An upwards polarization is established throughout the film during growth, resulting in the initially large ISHG signal. Once the growth is stopped, the top interface consolidates, and the

combined influence of both interfaces determines the resulting polarization. Specifically, the top interface promotes downwards polarization. The strong suppression of the net polarization as described above is a response that neither interface alone would generate; it is a remarkable manifestation of a combined interface effect. The immediate restoration of the initial ISHG signal upon resuming growth indicates a return to the state dictated by the bottom interface only. Accordingly, we identify the transient regime during growth as the "single-interface-contribution" regime and the persistent regime once the growth is stopped as the "combined-interface-contribution" regime.

Having revealed that competitive interfaces lead to a pronounced decrease of polarization, we now scrutinize the possibility of a polarization enhancement by cooperative interfaces. We choose $MnO_2$-terminated LSMO as the buffer, thus promoting downwards polarization at the bottom interface[5], while keeping the same strain and growth conditions as in the competitive case. During PTO deposition, the ISHG signal now evolves as follows. (i) Following a delayed onset of polarization, further detailed in Supplementary Note 2, the ISHG yield increases steadily with thickness. (Note that the bump in the SHG net intensity is caused by interference with surface-induced background SHG.) (ii) A giant tenfold enhancement of ISHG intensity is observed upon growth interruption. (iii) As before, the transient ISHG intensity is restored with the continuation of the growth.

The tenfold enhancement is an impressive manifestation of the difference between the action of a single (bottom) interface and the joint action of both (top and bottom) interfaces in the transient and persistent regime, respectively. It only appears in PTO films thinner than 40 u. c., see Fig. 2b, which emphasizes the role of interface proximity. Furthermore, whereas in the case of a single effective interface, upwards polarization exhibits the larger SHG yield and, thus, dipole moment (Fig. 1), the cooperative action of two interfaces promoting downwards polarization overrules this

preference (Fig. 2). We verified the giant cooperative polarization enhancement (ii) by STEM, finding a pervasive bulk-like OOP B-site displacement of about 16 pm (see Fig. 2c). The observed ISHG evolution (ii–iii) was verified on more than 40 samples, see Supplementary Note 3.

We next pinpoint the mechanism promoting pervasive downwards polarization at the top interface. We rule out the intrinsic chemistry of the PTO termination as a possible explanation because, according to ab initio calculations[22], both the PbO and the $TiO_2$ top termination favor an upwards polarization. Our tests with other ferroelectric perovskites rather point to non-stoichiometry as the likely origin because the striking polarization evolution (ii–iii) was only observed for materials with A-site volatility, i.e., PTO, Pb[$Zr_xTi_{1-x}$]$O_3$ (PZT), and $BiFeO_3$ (BFO), in contrast to $BaTiO_3$ (BTO)[4] (see Supplementary Note 3). A-site volatility is usually compensated by A-site-excess targets in pulsed laser deposition (PLD) to recover the stoichiometry in the bulk of the films. It can, however, induce non-stoichiometry at the film surface. In line with this, a non-stoichiometric phase of $Bi_2O_{3-x}$ at the surface has been shown to cause a local polarization change in BFO films[23,24]. Similarly, cationic vacancies were reported to affect the PTO polarization orientation[25]. Furthermore, the observed time for the reconstruction of polarization (ii) of about $10^2$ s matches the theoretically derived value in response to ion migration[26]. Most importantly, the post-growth surface-sensitive angle-resolved XPS data shown in Fig. 3a and energy dispersive X-ray analysis reveal increasing Pb content toward the surface of our PTO films for both the competitive and the cooperative interfaces (see details in Supplementary Note 4). The non-stoichiometry of PTO toward the surface is likely in the form of Pb adatoms. We note that other sources of non-stoichiometry such as Pb substitution of Ti atoms were not detected. In summary, we conclude that the Pb-rich top layer results in a positively charged top interface where it promotes a downwards polarization.

We use density functional theory (DFT) to verify that non-stoichiometry at the top interface determines the net polarization of the film in the persistent regime. We simulate the non-stoichiometry by introducing a Pb adatom at the top interface of a stoichiometric lattice of PTO[27–29]. We fix the direction of the dipole moment in the film at the bottom interface to either up or down to match with our experiments and allow the ions in the rest of the film to relax. Our calculations reveal that for an up-polarized bottom interface the resulting positive bound charge at the top interface is disfavored by the Pb adatom. In response, see Fig. 3b, the dipole moment is suppressed throughout the layer and reversed in the topmost u. c. to reduce the total energy of the film. In contrast, for a down-polarized bottom interface, the associated negative bound charge at the top interface is compensated by the positive charge of the bound Pb adatom (Fig. 3c). This results in a stabilization of the ferroelectric polarization, which remains bulk-like throughout the film.

We thus arrive at the following scenario for the growth dynamics of our PTO films. During growth (transient regime in Fig. 3d, e), epitaxial stabilization[30] promotes stoichiometry in the PTO films. The film polarization follows the state set at the bottom interface by the respective LSMO termination. Once the growth is stopped, the excess Pb from the Pb-rich target material (see "Methods") diffuses toward the film surface, which results in accumulation of positive charges at the top interface. In response, the polarization of the film is suppressed or enhanced (persistent regime in Fig. 3d, e).

We then investigate how the interface-induced polarization suppression and enhancement affect the electric-field poling of the PTO films. The results of the time-dependent switching behavior are shown in Fig. 4. In both interface configurations, the polarization can be locally reversed by the voltage applied to a

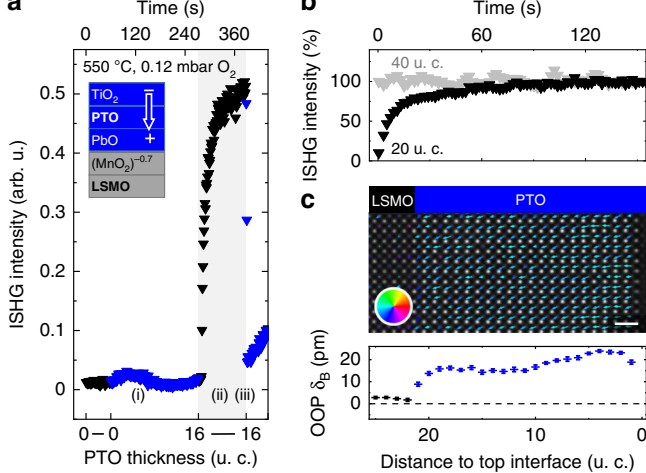

**Fig. 2 Polarization enhancement from cooperative interfaces. a** ISHG signal during ongoing (filled blue symbols, 0–16 u. c.) and halted (filled black symbols) growth. The inset shows the chemistries of interfacial planes at the PTO|LSMO interface and the polarization direction set by the bottom interface. **b** Time-dependent ISHG signal after interrupting deposition, normalized to the maximum value = 100. **c** Post-deposition STEM map of the dipole moments. The arrows show the direction (color wheel) and amplitude (arrow length) of the dipole moments (top). Scale bar is 1 nm. The OOP B-site displacement ($\delta_B$) of the persistent state is mapped throughout the PTO|LSMO bilayer (bottom). Black symbols refer to LSMO, and blue symbols refer to downwards polarization in PTO. The error bars are the standard error of the mean.

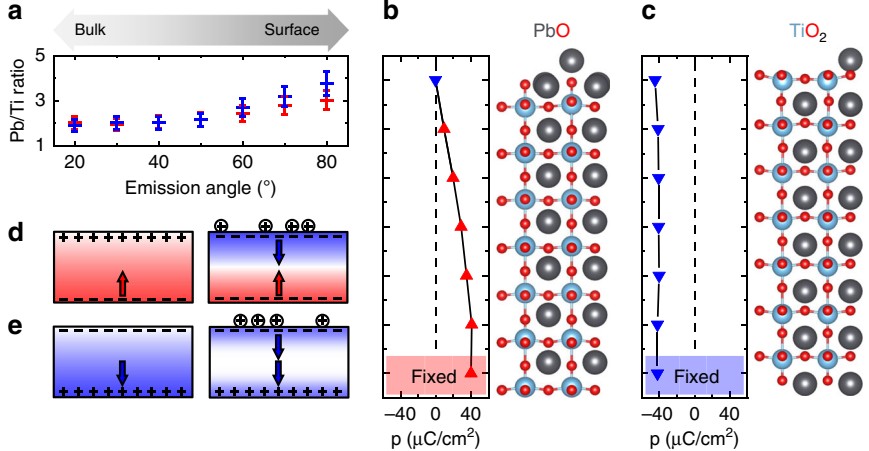

**Fig. 3 Competitive and cooperative configurations explored by XPS and DFT. a** Pb/Ti atomic ratio at XPS emission angles (20–80)°, higher angles displaying a higher sensitivity to the sample surface lower angles. Red and blue symbols correspond to the samples with suppressed and enhanced polarization, respectively. The error bar is based on the estimated experimental accuracy (see "Methods"). **b, c** Density-functional calculations of dipole moments (p) perpendicular to the bottom interface in PTO heterostructures with a Pb adatom on the surface. The polarization direction is imposed at the bottom interface to match with our experiments by fixing the dipole moment of the first u. c. in the calculation (marked as "fixed"). The atomic positions in heterostructures with (**b**) PbO and (**c**) TiO$_2$ top-interface termination and their ionic positions are depicted on the right-hand side of the graph. **d, e** Sketch of the competitive and cooperative interfaces for the transient (left) "single-interface" and the persistent (right) "combined-interface" regime. The favored polarization direction at each interface is indicated with arrows and the resulting bound charges are represented with "+" and "−" signs. The additional positive charges at the top interface introduced to emulate the non-stoichiometry are shown as "⊕".

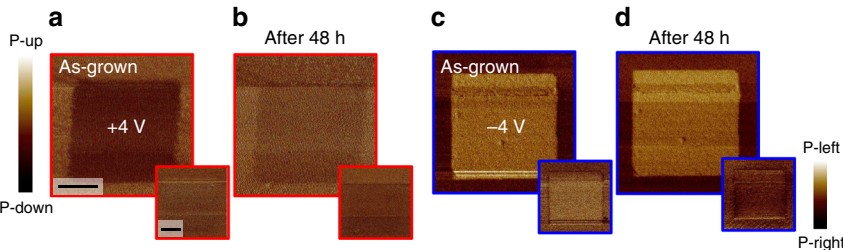

**Fig. 4 Stability of the electric-field-induced polarization switch over time. a–d** PFM out-of-plane (in-phase, main panel) and in-plane (quadrature, inset) data immediately after the poling (±4 V) and 48 h later for **a, b** competitive and **c, d** cooperative interface configurations. PTO films are 20 u. c. thick and scale bars are 1 µm.

piezoresponse-force-microscopy (PFM) tip. In the PTO films with competitive interfaces, the electric-field-induced downwards polarization vanishes after 48 h (see Fig. 4a, b). In contrast, in the PTO films with cooperative interfaces, a stable upwards polarization is sustained over the same time period (see Fig. 4c, d). This correlates with our ISHG and STEM data, indicating a robust ferroelectric response when cooperative interfaces are at work.

Finally, we test the robustness of the competitive and the cooperative polarization state by capping the PTO films with 18 u. c. of STO. STO isolates PTO from the charge-screening oxygen-rich growth atmosphere and thus enhances the depolarizing field, promoting a multidomain breakdown[4,31]. For the suppressed polarization state, this breakdown is indeed observed, as confirmed macroscopically and microscopically by the SHG and STEM measurements in Fig. 5a, c. For the enhanced polarization state, however, the single-domain state is retained, as seen in Fig. 5b, d, e. Although one might assume the state with the higher net polarization to be more prone to energy minimization through multidomain breakdown, it is, in fact, more robust against STO capping.

A possible explanation for this striking behavior is provided by the STEM data in Fig. 5d. We find a sharp dipole-moment discontinuity across the epitaxial PTO|STO interface, which is revealed by the absence of B-site displacements in the STO layer despite the bulk-like B-site displacements in the PTO layer. We conclude that the remnant positive charges from non-stoichiometry at the PTO top-interface efficiently screen the negative bound charges of the enhanced polarization state at the PTO|STO interface, hence preventing multidomain breakdown. In the case of the suppressed polarization state, on the other hand, the positive charges induced by the non-stoichiometry cannot screen the positive bound charges associated with the oppositely oriented polarization, and multidomain breakdown occurs (see Supplementary Note 4).

## Discussion

Hence we have shown how, in the ultrathin regime, the proximity and combined influence of both interfaces in a thin epitaxial oxide film determine its properties. We demonstrate drastic differences between competitive and cooperative interface configurations and to the action of a single interface. Specifically, in ferroelectric PTO, cooperative interfaces stabilize a bulk polarization state down to the ultrathin regime of about 10 u. c. and protect it against depolarizing-field-induced multidomain breakdown. Selective access to the interfaces and their correlation is enabled by a unique combination of ISHG, STEM, and XPS measurements. We expect our findings to be relevant for nanoscale oxide thin films in which interfacial chemistry and charge imbalance at large are important, beyond the

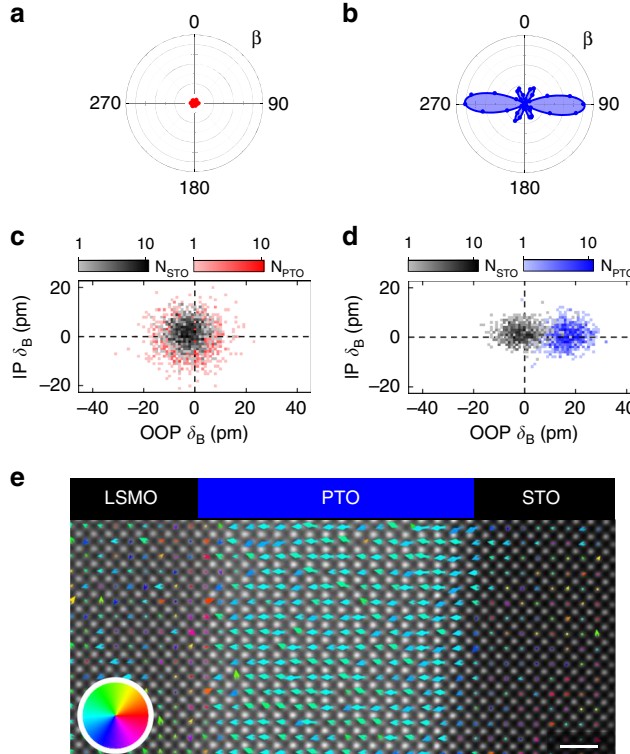

**Fig. 5 Stability of net polarization in a STO|PTO heterostructure.**
**a**, **b** SHG anisotropy plots of STO|PTO bilayers after cooling to room temperature, shown on the same intensity scale. Data were obtained by simultaneously varying the angle $\beta$ of the polarization of the incident fundamental and of the detected SHG light at 1200 and 600 nm, respectively. The simulations, see "Methods," are plotted as continuous lines. Note that for the competitive interfaces (**a**), a measurable signal is not detected. **c**, **d** Logarithmic histogram plots showing the number of occurrences $N_{STO}$ (black) and $N_{PTO}$ of OOP and in-plane (IP) B-site displacement ($\delta_B$) measured in post-deposition STEM images for the (**c**) competitive (red) and the (**d**) cooperative (blue) interface configuration. The zero net value of B-site displacement in the OOP direction of the competitive case in the multidomain state is the result of polarization averaging across the thickness of the lamella. **e** Post-deposition STEM map of PTO dipole moments after STO capping in the cooperative configuration. The arrows show the direction (color wheel) and amplitude (arrow length) of the dipole moments. Scale bar is 1 nm.

specific A-site volatile proper ferroelectrics investigated here. Bi-interfacial control of ultrathin films can thus become a key factor in bypassing present limitations of oxide-electronic functionalities and devices[32,33].

## Methods

**Thin-film deposition.** The thin films and heterostructures were grown on TiO$_2$-terminated STO (001) substrates by PLD using a KrF excimer laser at 248 nm. The fluence of the laser, its repetition rate, substrate temperature, and growth pressure for individual layers were as follows: SrRuO$_3$ (SRO): 0.9 Jcm$^{-2}$, 2 Hz, 700 °C, 0.1 mbar O$_2$; LSMO: 0.9 Jcm$^{-2}$, 1 Hz, 700 °C, 0.15 mbar O$_2$; PTO: 1.15 Jcm$^{-2}$, 4 Hz, 550 °C, 0.12 mbar O$_2$; and STO: 1.15 Jcm$^{-2}$, 2 Hz, 550 °C, 0.12 mbar O$_2$. The PTO target was Pb-enriched (Pb$_{1.2}$TiO$_3$) and the other targets were stoichiometric. The thickness of the thin films was monitored using RHEED during growth. The thin-film topography and PFM were performed using a Bruker Multimode 8 atomic force microscope.

**In situ second harmonic generation (ISHG).** The incident light was generated from a Ti:Sapphire laser light with a pulse duration of 45 fs, a repetition rate of 1 kHz, and a wavelength of 800 nm, which was converted using an optical parametric amplifier to light with a wavelength of 1200 nm. This incident beam was focused onto the sample in the thin-film growth environment in reflection at 45° with a pulse energy of 30 μJ and a spot size 250 μm in diameter. The optical SHG signal was generated at 600 nm and detected using a monochromator and a photomultiplier system.

**Analysis of SHG signal.** The SHG process is expressed by the equation $P_i(2\omega) = \epsilon_0 \chi^{(2)}_{ijk} E_j(\omega) E_k(\omega)$, where $E_{j,k}(\omega)$ and $P_i(2\omega)$ are the electric-field components of the incident light and of the frequency-doubled polarization, respectively. The point-group symmetry of a compound determines the set of its tensor components $\chi^{(2)}_{ijk} \neq 0$. The anisotropy simulations of the tetragonal $4mm$ group were consistent with the SHG data when the components $\chi^{(2)}_{xzx}$, $\chi^{(2)}_{zxx}$, and $\chi^{(2)}_{zzz}$ were fitted as nonzero.

**Scanning transmission electron microscopy (STEM).** Cross-sectional specimens for the STEM analysis were prepared by mechanical polishing using a tripod polisher followed by argon ion milling using a Fishione ion miller model 1050 operated at 3 kV until electron transparency. A FEI Titan Themis equipped with a probe CEOS DCOR spherical aberration corrector and ChemiSTEM technology operated at 300 kV was used for HAADF-STEM imaging and EDX spectroscopy. The atomic-resolution HAADF-STEM images were acquired setting a probe semi-convergence angle of 25 mrad in combination with an annular semi-detection range of the annular dark-field detector of 66–200 mrad. To correct for the scan distortions, time series consisting of 10 frames (2048 × 2048 pixels) were acquired and averaged by rigid and nonrigid registration by means of the Smart Align software[34]. A Gaussian filter was applied, followed by a custom-developed Python code for blind probe deconvolution assuming a Gaussian distribution as the initial probe to reduce the spread of the atomic columns. Subsequently, fitting of the atomic columns was performed using the Python library Atomap[35] and the local dipole moment was calculated by measuring the polar displacement of the B position in the image plane from the center of mass of its four nearest A neighbors. The dipole moment is plotted in the HAADF-STEM images opposite to the B-site displacement ($\delta_B$).

**X-ray photoelectron spectroscopy (XPS).** X-ray photoelectron spectra were acquired using a PHI Quantera$^{SXM}$ (ULVAC-PHI, Chanhassen, MN, U.S.A.) spectrometer. The films were mounted on standard PHI platen and analyzed using a monochromatic Al K$_\alpha$ source ($h\nu = 1486.6$ eV at 24.5 W); the beam diameter was of 100 μm. The analyzer was operated in the fixed analyzer transmission mode. A pass energy of 69 eV and a step size of 0.125 eV were used to obtain the high-resolution spectra of C 1s, O 1s, Ti 2p, and Pb 4f. The linearity of the energy scale was checked according to ISO15472: 2010 (confirmed in 2015) and the accuracy of the binding-energy value for each element was found to be 0.1 eV. The binding energies were referred to adventitious aliphatic carbon signal taken at 285.0 eV. The angle-resolved spectra were acquired using the special sample holder designed by ULVAC-PHI for these measurements, and the calibration for the eucentric tilt was carried out before each spectral acquisition so that the spectra were collected on the samples by maintaining the analysis area always on the focal point of the x-ray source and of the analyzer at all emission angles. The estimated experimental accuracy of 10% is primarily a result of: signal-to-noise ratio, peak intensity, accuracy of relative sensitivity factors, and correction for electron transmission function. During the analysis, the residual pressure in the main chamber was measured to be around $10^{-7}$ Pa. The spectra were processed using CASAXPS software (version 2.3.15, Casa Software Ltd.).

**Ab initio calculations.** DFT calculations of PTO thin films, both without buffer and on LSMO were carried out with VASP[36] using the PBEsol[37] functional, which gives a good metal/insulator band alignment at the interface, with the Fermi level of the metal falling within the PTO gap[38]. The alloying in LSMO was simulated using the virtual crystal approximation[39,40]. Core electrons were replaced by projector augmented wave potentials[41], and the valence states were expanded in plane waves with a cutoff energy of 500 eV. A Monkhorst-Pack grid of 3 × 3 × 1 $k$-points was used for all thin films with a surface area of 2 × 2 u. c. A 14-layer-thick (1 × 1) PTO (001) slab, consisting of 7 u. c. was found to be sufficient to converge the surface energy. The in-plane and out-of-plane lattice parameters were set to those of the STO substrate ($a = 3.905$ Å) and the PTO bulk value ($c = 4.11$ Å), respectively. In all cases, the atoms of the four bottom layers of the PTO were fixed to the calculated bulk values, while the rest of the system was allowed to relax.

## Data availability
The data that support the findings of this study are available from the corresponding authors upon request.

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

## Acknowledgements

M.T. acknowledges the Swiss National Science Foundation under Project No. 200021-188414. N.S., M.T. and M.F. acknowledge support by the EU European Research Council under Advanced Grant Program No. 694955-INSEETO. M.F. acknowledges support by the Swiss National Science Foundation under Project No. 200021-178825. N.S. and M.T. thank Gabriele De Luca for fruitful discussions and Elzbieta Gradauskaite and Martin Sarott for experimental assistance. C.G. is supported by the European Union's Horizon 2020 research and innovation programme under the Marie Skłodowska-Curie Grant Agreement No. 744027. C.G.'s computational work was supported by a grant from the Swiss National Supercomputing Centre (CSCS) under project ID s870. N.S., C.G., A.R., N.A.S., M.F., and M.T. acknowledge support from ETH Zurich. A.V., M.C., and M.D.R. acknowledge support by the Swiss National Science Foundation under Project No. 200021-175926. A.R. thanks technical support from Giovanni Cossu.

## Author contributions

All authors discussed the results. N.S., M.T., and M.F. wrote the manuscript. N.S. performed the thin-film growth, PFM, ISHG measurements and structural analysis. C.G., R.H., and N.A.S. performed the DFT calculations. A.V., M.C. and M.D.R. carried out the STEM investigations. A.R. supervised, processed, and interpreted the XPS data. M.T. designed the experiment and supervised the work jointly with M.F.

## Competing interests

The authors declare no competing interests.
