## [Peer Review File · Nature Communications]

REVIEWER COMMENTS

Reviewer #1 (Remarks to the Author):

In ferroelectrics, a major obstacle to stabilizing the electric polarization in ultrathin layers is uncompensated bound charge at the interfaces. The authors observed drastic differences between the influence of a single interface and the competition and cooperation of two interfaces in PbTiO₃. This is achieved by observing the spontaneous polarization right when it is formed — during assembly of the film in the growth chamber using an in situ second harmonic generation (ISHG) technique. They found that a robust polarization state with giant polarization enhancement can be stabilized by tailoring the interface chemistry towards a cooperative configuration. They further explained the observation through scanning transmission electron microscopy (STEM), angle-resolved X-ray photoelectron spectroscopy (XPS) and first-principles calculations. These results are interesting and I would like to recommend the publication of this work in Nature Communications after the authors consider the following comments:

(1) The authors wrote “Our tests with other ferroelectric perovskites rather point to non-stoichiometry as the likely origin because the striking polarization evolution (ii-iii) was only observed for materials with A-site volatility, i. e. PTO, Pb[Zr_xTi_{1-x}]O₃ and BiFeO₃ (BFO), but not BaTiO₃.” Can the authors discuss the results on other ferroelectric perovskites in more detail (maybe in SM) so as to make the conclusion more general?

(2) The authors proposed that “The Pb-excess is thus the most probable mechanism promoting downwards polarization at the top interface.” However, in the DFT calculations, the authors simulated the non-stoichiometric positive charges by introducing hydrogen ions at the top interface of a stoichiometric lattice. Can the authors model the system in a more direct way, i.e., introducing Pb ions at the top interface?

(3) The authors found that one can use interface cooperation to stabilize a robust polarization state with giant polarization enhancement. Can the electric polarization be fully switched by an electric field as the polarization depends on the interfaces?

Reviewer #2 (Remarks to the Author):

This paper presents a study on the stability of ferroelectric polarization in very thin PbTiO₃ films, and specifically the effect of interface chemical termination at the lower electrode. Nanoscale ferroelectrics are quite interesting to study for both a fundamental insight and also for potential nanoscale devices, so the contribution is timely and interesting to a wide readership. Combining in-situ second harmonic generation optical measurements with pulsed laser deposition thin film growth and other ex-situ experimental methods (transmission electron microscopy, SHG, PFM, XPS), the authors show that the stable polarization direction in PbTiO₃ thin films grown on SrTiO₃ depends on whether the lower interface or upper interface work in concert or antagonistically, and this defines the final polarization state. The results are quite interesting, and I think would be appropriate for the audience of Nature Communications. However, I feel that the paper lacks clarity of presentation with some of the ideas not well expressed. Therefore, it could be significantly improved. Below is a list of various issues and questions which should be carefully considered and acted upon before the paper is suitable for publication.

1.

The argument about Pb-rich surface of their films is unconvincing. First, since Pb is volatile, one would expect that the surface if anything should be Pb deficient. On lines 110 to 114 authors reason that a Pb excess at the surface of the films could be arising from substitution of Ti atoms by Pb. But is there any evidence of such antisite defects by the STEM? Using the HAADF technique, presumably they are

sensitive to the z contrast and since the atomic numbers of Ti and Pb are so different, a clear contrast difference should be observed in the STEM images if their hypothesis is correct. Please comment. Some experimental method other than the XPS could be used to confirm this hypothesis (maybe elemental mapping using EELS or EDS within the TEM measurements?). Also, Ref 10 in the supplementary talks about XPS results depending strongly on the polarization state at the surface, in addition to significant Pb-deficiency regardless of the polarization direction; however, these points appear not to have been taken into account in the present manuscript's reasoning. Why is this?

2.

The layout of the paper could be rearranged to help readability. Perhaps too much information is put into the supplementary making it inefficient to switch back and forth between the documents. For example, at line 61, the authors write "First, dipole moments next to the top interface are reversed with respect to the direction set by the bottom interface" but for the reader to find this evidence s/he has to go to the supplementary information. It would be better to put the TEM polarization map in the main figure to help readability. I understand that the tetragonality of the unit cell is contained in Fig. 1c, but this does not reflect the strange polarization behaviour that the authors describe in the text. Also, why in Figure 1 does the ISHG intensity have no units but then in the supplementary there are arbitrary units (which are later compared to each other, see comment 4 below)?

I encourage the authors to revisit the layout of their paper to try to improve the readability of the manuscript and to avoid the need to switch back and forth from SI.

3.

Figure 1a: What happens if the growth is interrupted for longer? It appears that the black data points are still heading downwards, so "ISHG intensity gradually decreases to a new stable value" is perhaps misleading. If the growth is interrupted for longer, does the ISHG intensity continue to decrease? And if so, why? There are some handy wavy explanations for this but for me this is not super convincing.

4.

Line 89: "The tenfold enhancement..." but the ISHG axis has no units so the reader cannot easily see this tenfold enhancement.

Also regarding the ISHG intensity, which is measured in arbitrary units. It is mentioned in the SI that "The final ISHG intensity of the competitive interfaces is four times lower than the final ISHG intensity of the cooperative interfaces." What are the factors that dictate the ISHG signal? In the main paper it is explained that the signal is proportional to the square of the polarization but in this case is there some effect due to the physical volume of thin film being probed? If so, the comparison in Figure SI(c,d) is a bit misleading as the film thicknesses are not the same? Also, regarding that figure, it might be more intuitive to plot the ISHG intensity with the same scale for both to improve clarity.

5.

Line 99-100: "We rule out PTO termination as a possible explanation because both the PbO and the TiO₂ top termination would favor upwards polarization²²" with reference 22 being a paper "under review". Here it would be helpful to the reader to add at least a couple of sentences to explain this reasoning (or better, another relevant reference if it exists).

6.

Line 96: "The observed ISHG evolution (ii-iii) was verified for more than 40 samples and for PTO layers up to 35 u. c." This statement is good, but it might be nice to have a supplementary note giving some other examples. For instance, a few representative figures or even a table summarizing these results from 40 samples. As a bare statement like this, it looks like an afterthought.

7.

It would be interesting to see if there is a difference in the retention of the polarization state for the different types of interfaces (antagonistic vs. cooperative) – do the authors have some PFM data after e.g. a few days for both the images shown in Supp Figure 2?

8.

What is the origin of the apparently in plane rotation of the PTO in Figure S4d? If PTO is uniaxial, then we would expect an (almost) out of plane polarization vector at the unit cell level. Especially for the PTO with cooperative interfaces (blue) there seems to be some rotation of the P vector.

9.

Why are the experiments not conducted in exactly the same way for cooperative and competitive interfaces? In Figure S1c the first 'phase' of growth of PTO is 8 unit cells while for S1d it is 16 unit cells. This is also evident in the Figure S3(a,b). The growth protocols are not the same for both, which makes the interpretation a bit confusing. Maybe it's better to compare exactly the same growth protocols for both types of sample so that there is an "apples with apples" comparison?

10.

Figure 4e. Where is the corresponding STEM image for the competitive interfaces case?

Reviewer #3 (Remarks to the Author):

In this paper, the authors report on the effects of interface proximity on polarization in ultrathin ferroelectrics. They study the polarization in PbTiO₃ thin films during PLD growth using in-situ harmonic generation. This technique allows them to follow the development of polarization during growth, i.e. as a function of film thickness, but also when they turn the laser beam off and stop deposition, and once they add a top electrode or capping layer. The technique is quite impressive by itself, and it is really interesting to see how the polarization develops during the growth and how sensitive it is to the presence of the plume and to the addition of a top layer. They also compare what happens when they use different bottom electrode terminations, favouring one polarisation over the other at the bottom interface. The whole work has been repeated on many samples (over 40 samples) and additional characterisations have been performed using STEM and XPS. The conclusions of the paper are also supported by first-principles calculations. The paper is extremely well written, and it is clear that all the measurements and calculations have been performed very carefully.

The authors specifically study two different configurations, that they call "competitive" (where the up-polarization is favoured at the bottom interface while the down-polarization is favoured at the top) and "cooperative" (where both interfaces favour the down-polarization). This is achieved by engineering the termination of the bottom interface (LSMO) to favour one polarization orientation or the other. In the "competitive" configuration, the polarization reverses along the film thickness, starting by pointing up at the bottom interface, and pointing down at the top interface. In the "cooperative" configuration, the polarization (pointing down everywhere along the film thickness) is enhanced.

The authors also monitor the iSHG response when the growth is interrupted. For the measurements performed on the sample in the "competitive" configuration, the authors conclude that they identified "a transient "single-interface" regime during growth and a persistent "combined-interface" regime once the growth is stopped" (l.76-79), from the fact the iSHG intensity increases during the growth and then decreases as soon as the growth is interrupted. I do not agree with these conclusions. I believe that during the growth, in the presence of the plume, the top layer is affected in a way that

favours the up polarization, as discussed indeed in the supplementaries (Section 1). This is therefore not a transition from a "single-interface" regime to a "combined-interface" regime. The system always has two interfaces that clearly play a role on the polarization configuration. It is just the conditions at the top interface that are modified while the growth is on or off.

The authors also conclude on the "interaction" between the two interfaces (see for example 1.27) and the importance of interfaces "proximity" (see title). With this also I disagree. It is clear from their measurements that both interfaces (bottom and top) have an importance in the sample polarization configuration. But I do not see any hint of "interaction" between the two or the importance of their "proximity". I believe the authors should just reformulate these affirmations and maybe also adapt the title. I think that in the title they should emphasise the fact that they monitor the polarization in-situ during the growth.

Once these two major issues are addressed, I believe this paper deserves publication.

The DFT calculations should be explained a bit more in details in the main text. It is not clear how the polarization gradient is screened in the DFT and if there is a gradient in stoichiometry. Positive charges at the top interface are added to emulate the non-stoichiometry: this is briefly mentioned in the caption of figure 3 but should be explained in more details in the text.

Below are additionally minor details that can be addressed:

- Figures 1(a) and 2(b) - I suggest to put the Time as the lower axis and indicate the PTO thickness at the top.

- Also from Figure 1 and 2, the authors say the "the time axis for this growth protocol reveals relaxation of the polarization on the order of 10^2 s" - this is not so clear from the graph (also in supplementary Figure 1)

- Figure 3(a): the scale for the Pb/Ti ratio varies between 1 and 5, and the points vary from 2 to 4. This clearly must be wrong.

- Figure 4(c): this corresponds to the "competitive" configuration where part of the sample has a polarization pointing down and part of the sample has a polarization pointing up. Why then are the OOP values scattered around 0 for PTO, rather than having two clusters with positive and negative values?

- Supplementary section 2: piezoresponse force microscopy: the authors should describe the pattern that was written and mention what are the "virgin" regions. They should also comment on the stability of the written regions. Did they observe some back-switching? Could they also add the corresponding amplitude images? In the figure caption, the authors mixed "b" and "c".

- Supplementary Section 3: the minus signs are missing in the formula for the Schottky barriers: $E_c - E_f$ and $E_f - E_v$ (instead of $E_c E_f$ and $E_f E_v$).

- Supplementary Section: I don't agree when the authors write that the tetragonality according to STEM is uniform throughout both types of samples - it clearly isn't uniform in figures a and b.

Response to Reviewers: Effects of interface proximity on polarization in ultrathin ferroelectrics (NCOMMS-20-24592)

First of all, we would like to thank the Reviewers for their careful reading of the manuscript and their constructive comments. Our point-by-point response to the Reviewers is shown in black. We have revised the main text and the Supplementary Information file to address all questions.

Reviewer #1 (Remarks to the Author):

1.1) In ferroelectrics, a major obstacle to stabilizing the electric polarization in ultrathin layers is uncompensated bound charge at the interfaces. The authors observed drastic differences between the influence of a single interface and the competition and cooperation of two interfaces in PbTiO_3 . This is achieved by observing the spontaneous polarization right when it is formed — during assembly of the film in the growth chamber using an in situ second harmonic generation (ISHG) technique. They found that a robust polarization state with giant polarization enhancement can be stabilized by tailoring the interface chemistry towards a cooperative configuration. They further explained the observation through scanning transmission electron microscopy (STEM), angle-resolved X-ray photoelectron spectroscopy (XPS) and first-principles calculations. These results are interesting and I would like to recommend the publication of this work in Nature Communications after the authors consider the following comments:

Answer 1.1:

We appreciate the favorable opinion of Reviewer #1 on the suitability of publishing our manuscript in Nature Communications.

1.2) The authors wrote “Our tests with other ferroelectric perovskites rather point to non-stoichiometry as the likely origin because the striking polarization evolution (ii-iii) was only observed for materials with A-site volatility, i. e. PTO , $\text{Pb}[\text{Zr}_x\text{Ti}_{1-x}]\text{O}_3$ and BiFeO_3 (BFO), but not BaTiO_3 .” Can the authors discuss the results on other ferroelectric perovskites in more detail (maybe in SM) so as to make the conclusion more general?

Answer 1.2:

We now added the ISHG intensity during growth and during growth interruption for BaTiO_3 , $\text{Pb}[\text{Zr}_x\text{Ti}_{1-x}]\text{O}_3$ and BiFeO_3 thin films in Supplementary Section 3 to strengthen the generality of our conclusions as suggested.

1.3) The authors proposed that “The Pb-excess is thus the most probable mechanism promoting downwards polarization at the top interface.” However, in the DFT calculations, the authors simulated the non-stoichiometric positive charges by introducing hydrogen ions at the top interface of a stoichiometric lattice. Can the authors model the system in a more direct way, i.e., introducing Pb ions at the top interface?

Answer 1.3:

We followed the Reviewer#1's suggestion and performed additional calculations with Pb adatoms. We edited the main text in lines 122-130 accordingly. The results show that the

suppression and enhancement of polarization upon binding of plus-charged ions are qualitatively independent of the exact chemistry of the ion (H^+ or Pb^{2+}) and are thus mainly of electrostatic origin.

1.4) The authors found that one can use interface cooperation to stabilize a robust polarization state with giant polarization enhancement. Can the electric polarization be fully switched by an electric field as the polarization depends on the interfaces?

Answer 1.4:

Yes, in both interface configurations, the electric polarization can be fully switched by an electric field. We now include results of the switching retention experiments and the corresponding discussion in Figure 4 and in the main text in lines 139-147. We observe superior room-temperature ferroelectric properties of the films grown in the configuration with interface cooperation.

Reviewer #2 (Remarks to the Author):

2.1) This paper presents a study on the stability of ferroelectric polarization in very thin $PbTiO_3$ films, and specifically the effect of interface chemical termination at the lower electrode. Nanoscale ferroelectrics are quite interesting to study for both a fundamental insight and also for potential nanoscale devices, so the contribution is timely and interesting to a wide readership. Combining in-situ second harmonic generation optical measurements with pulsed laser deposition thin film growth and other ex-situ experimental methods (transmission electron microscopy, SHG, PFM, XPS), the authors show that the stable polarization direction in $PbTiO_3$ thin films grown on $SrTiO_3$ depends on whether the lower interface or upper interface work in concert or antagonistically, and this defines the final polarization state. The results are quite interesting, and I think would be appropriate for the audience of Nature Communications. However, I feel that the paper lacks clarity of presentation with some of the ideas not well expressed. Therefore, it could be significantly improved. Below is a list of various issues and questions which should be carefully considered and acted upon before the paper is suitable for publication.

Answer 2.1:

We thank Reviewer #2 for acknowledging the general interest of the manuscript for publication in Nature Communications. We improved the clarity of the manuscript on the points raised by the reviewer as discussed below.

2.2) The argument about Pb-rich surface of their films is unconvincing. First, since Pb is volatile, one would expect that the surface if anything should be Pb deficient. On lines 110 to 114 authors reason that a Pb excess at the surface of the films could be arising from substitution of Ti atoms by Pb. But is there any evidence of such antisite defects by the STEM? Using the HAADF technique, presumably they are sensitive to the z contrast and since the atomic numbers of Ti and Pb are so different, a clear contrast difference should be observed in the STEM images if their hypothesis is correct. Please comment. Some experimental method other than the XPS could be used to confirm this hypothesis (maybe elemental mapping using EELS or EDS within the TEM measurements?).

Answer 2.2:

To support our argument of the Pb-rich surface, we now emphasize the use of a Pb-rich target (20% atomic excess) in the main text in lines 108-110 and refer to previous reports dealing with similar accumulation of the A-site volatile element at the surface of oxide films [Ref. 23 and 24: Xie, L. et al., *Adv. Mater.* **29**, 1701475 (2017) and Béa, H. et al. *Appl. Phys. Lett.* **87**, 072508 (2005)]. Furthermore, as recommended, we performed additional composition analysis of the PTO surface using energy dispersive X-ray (EDX). We clearly identify a Pb-rich surface layer in both interface configurations, shown in Supplementary Section 4 which is in line with our XPS investigation. Enlightened by this EDX-based observation, we now edited the main text in lines 115-120.

2.3) Also, Ref 10 in the supplementary talks about XPS results depending strongly on the polarization state at the surface, in addition to significant Pb-deficiency regardless of the polarization direction; however, these points appear not to have been taken into account in the present manuscript's reasoning. Why is this?

Answer 2.3:

We apologize for the confusion brought by referencing this work. In Ref. 10, the XPS was performed under different conditions (samples grown by off-axis magnetron sputtering in 180 mTorr oxygen/argon mixture, defect accumulation post-growth, XPS in ambient pressure) than ours (samples grown by PLD, Pb/Ti ratio set during the growth process, XPS in vacuum). Since the Pb excess is now further evidenced by EDX, we found that this Supplementary Section is no longer necessary and deleted it.

2.4) The layout of the paper could be rearranged to help readability. Perhaps too much information is put into the supplementary making it inefficient to switch back and forth between the documents. For example, at line 61, the authors write "First, dipole moments next to the top interface are reversed with respect to the direction set by the bottom interface" but for the reader to find this evidence s/he has to go to the supplementary information. It would be better to put the TEM polarization map in the main figure to help readability. I understand that the tetragonality of the unit cell is contained in Fig. 1c, but this does not reflect the strange polarization behaviour that the authors describe in the text.

Answer 2.4:

We now show the STEM polarization maps in the main text in Figures 1c and 2c. The tetragonality is depicted in Supplementary Section 4.

2.5) Also, why in Figure 1 does the ISHG intensity have no units but then in the supplementary there are arbitrary units (which are later compared to each other, see comment 4 below)? I encourage the authors to revisit the layout of their paper to try to improve the readability of the manuscript and to avoid the need to switch back and forth from SI.

Answer 2.5:

We edited Figures 1 and 2 in the main text to reduce the reference to SI and arbitrary units are now consistently given for the ISHG.

2.6) Figure 1a: What happens if the growth is interrupted for longer? It appears that the black data points are still heading downwards, so “ISHG intensity gradually decreases to a new stable value” is perhaps misleading. If the growth is interrupted for longer, does the ISHG intensity continue to decrease? And if so, why? There are some handy wavy explanations for this but for me this is not super convincing.

Answer 2.6:

The results of a longer growth interruption are now added in Figures 1b and 2b. We show the stabilization of the ISHG intensity at a new value after about 120 s.

2.7) Line 89: “The tenfold enhancement...” but the ISHG axis has no units so the reader cannot easily see this tenfold enhancement.

Answer 2.7:

This is indeed a glitch. We added the units to the ISHG intensity in Figures 1a and 2a in the main text.

2.8) Also regarding the ISHG intensity, which is measured in arbitrary units. It is mentioned in the SI that “The final ISHG intensity of the competitive interfaces is four times lower than the final ISHG intensity of the cooperative interfaces.” What are the factors that dictate the ISHG signal? In the main paper it is explained that the signal is proportional to the square of the polarization but in this case is there some effect due to the physical volume of thin film being probed? If so, the comparison in Figure SI(c,d) is a bit misleading as the film thicknesses are not the same? Also, regarding that figure, it might be more intuitive to plot the ISHG intensity with the same scale for both to improve clarity.

Answer 2.8:

The polarization-related SHG amplitude is proportional to $P \cdot t$, where P is the polarization and t is the thickness of the film. We added this information in the main text in lines 41-42. We now compare the ISHG intensity of two samples with the same thickness of 20 u. c. in Supplementary Section 1.

2.9) Line 99-100: “We rule out PTO termination as a possible explanation because both the PbO and the TiO₂ top termination would favor upwards polarization²²” with reference 22 being a paper “under review”. Here it would be helpful to the reader to add at least a couple of sentences to explain this reasoning (or better, another relevant reference if it exists).

Answer 2.9:

This statement refers to ab-initio calculations which reveal that both PbO and TiO₂ termination at the top interface favor upwards polarization. Since we observe that the top interface promotes downwards polarization instead, we can rule out that the top-interface termination plays a dominant role in setting the polarization at the top interface. We updated the reference 22 which is accepted in PNAS. We now added this explanation in the main text in lines 101-105.

2.10) Line 96: “The observed ISHG evolution (ii–iii) was verified for more than 40 samples and for PTO layers up to 35 u. c.” This statement is good, but it might be nice to have a supplementary note giving some other examples. For instance, a few representative figures or

even a table summarizing these results from 40 samples. As a bare statement like this, it looks like an afterthought.

Answer 2.10:

As suggested by Reviewer #2, we now created Supplementary Section 3 showing the reproducibility and generality of our experiments on several samples and other ferroelectric thin-film systems such as BiFeO₃ and PZT.

2.11) It would be interesting to see if there is a difference in the retention of the polarization state for the different types of interfaces (antagonistic vs. cooperative) – do the authors have some PFM data after e.g. a few days for both the images shown in Supp Figure 2?

Answer 2.11:

We added an analysis of the time-dependence of the switching retention to the main text in lines 139-147 and Figure 4. Please refer to Answer 1.4.

2.12) What is the origin of the apparently in plane rotation of the PTO in Figure S4d? If PTO is uniaxial, then we would expect an (almost) out of plane polarization vector at the unit cell level. Especially for the PTO with cooperative interfaces (blue) there seems to be some rotation of the P vector.

Answer 2.12:

This is an artefact caused by a residual sample tilt with respect to the incoming electron beam. This sample tilt results in “artificial” polarization. We performed a new measurement series and now show data with no in-plane rotations.

2.13) Why are the experiments not conducted in exactly the same way for cooperative and competitive interfaces? In Figure S1c the first ‘phase’ of growth of PTO is 8 unit cells while for S1d it is 16 unit cells. This is also evident in the Figure S3(a,b). The growth protocols are not the same for both, which makes the interpretation a bit confusing. Maybe it’s better to compare exactly the same growth protocols for both types of sample so that there is an “apples with apples” comparison?

Answer 2.13:

Our initial experiments revealed that the behavior in terms of ISHG suppression and enhancement is independent of the thickness at which the growth is interrupted and the duration of the growth interruption for the regime of interface proximity, see Supplementary Section 3. We nevertheless edited Supplementary Figure 1 to show the ISHG data for both interface configurations with exactly the same growth protocols, as requested.

2.14) Figure 4e. Where is the corresponding STEM image for the competitive interfaces case?

Answer 2.14:

We now include the corresponding STEM image for the competitive configuration in Supplementary Section 4.

Reviewer #3 (Remarks to the Author):

3.1) In this paper, the authors report on the effects of interface proximity on polarization in ultrathin ferroelectrics. They study the polarization in PbTiO₃ thin films during PLD growth using in-situ harmonic generation. This technique allows them to follow the development of polarization during growth, i.e. as a function of film thickness, but also when they turn the laser beam off and stop deposition, and once they add a top electrode or capping layer. The technique is quite impressive by itself, and it is really interesting to see how the polarization develops during the growth and how sensitive it is to the presence of the plume and to the addition of a top layer. They also compare what happens when they use different bottom electrode terminations, favouring one polarisation over the other at the bottom interface. The whole work has been repeated on many samples (over 40 samples) and additional characterisations have been performed using STEM and XPS. The conclusions of the paper are also supported by first-principles calculations. The paper is extremely well written, and it is clear that all the measurements and calculations have been performed very carefully.

The authors specifically study two different configurations, that they call "competitive" (where the up-polarization is favoured at the bottom interface while the down-polarization is favoured at the top) and "cooperative" (where both interfaces favour the down-polarization). This is achieved by engineering the termination of the bottom interface (LSMO) to favour one polarization orientation or the other. In the "competitive" configuration, the polarization reverses along the film thickness, starting by pointing up at the bottom interface, and pointing down at the top interface. In the "cooperative" configuration, the polarization (pointing down everywhere along the film thickness) is enhanced.

Answer 3.1:

We thank Reviewer #3 on the expressed appreciation of our work and fully agree with the reviewer's summary.

3.2) The authors also monitor the iSHG response when the growth is interrupted. For the measurements performed on the sample in the "competitive" configuration, the authors conclude that they identified "a transient "single-interface" regime during growth and a persistent "combined-interface" regime once the growth is stopped" (1.76-79), from the fact the iSHG intensity increases during the growth and then decreases as soon as the growth is interrupted. I do not agree with these conclusions. I believe that during the growth, in the presence of the plume, the top layer is affected in a way that favours the up polarization, as discussed indeed in the supplementaries (Section 1).

Answer 3.2:

We indeed considered the influence of the PLD plume, but ruled it out for the following reason: the front of the plume is rich in heavier ions, in our case cations, and is therefore positively charged [Ojeda-G-P, et al. *Adv. Mater. Interfaces* **5**, 1701062 (2019)]. Such a positively charged environment would preferably screen negatively charged surfaces (down-polarized) during the growth and create a suppression of polarization once the growth is interrupted. Our observations, in contrast, reveal a delayed onset of the polarization and the polarization enhancement once the growth is stopped for the downwards-polarized films.

We now expanded the discussion in Supplementary Section 1.

3.3) This is therefore not a transition from a "single-interface" regime to a "combined-interface" regime. The system always has two interfaces that clearly play a role on the polarization configuration. It is just the conditions at the top interface that are modified while the growth is on or off.

Answer 3.3:

We fully agree that the system always has two interfaces. Our terminology of a "single interface" was chosen because of the relatively slow impact of the top interface on the polarization state of the films (ca. 120 s independent of the polarization direction in Figures 1b and 2b) with respect to the growth process at 4Hz. Hence, only the bottom interface effectively contributes, and the top interface is non-contributing to the polarization of the PTO film during the growth. We now realize that this terminology may nevertheless sound misleading and therefore we added the explanation above in the main text in lines 70-71 and refer to the two regimes as "single-interface-contribution" and "combined-interface-contribution".

3.4) The authors also conclude on the "interaction" between the two interfaces (see for example 1.27) and the importance of interfaces "proximity" (see title). With this also I disagree. It is clear from their measurements that both interfaces (bottom and top) have an importance in the sample polarization configuration. But I do not see any hint of "interaction" between the two or the importance of their "proximity". I believe the authors should just reformulate these affirmations and maybe also adapt the title. I think that in the title they should emphasise the fact that they monitor the polarization in-situ during the growth.

Answer 3.4:

We used the word "interaction" here in a broad sense to indicate that both interfaces are important. We now realize that this terminology can be misleading and replaced this with "joint effects" in the main text in line 28. We apologize for the lack of information in the previous version of the manuscript regarding interface proximity. We now show that the suppression and enhancement occur only in the case of interface proximity. We compare in Figures 1b and 2b the polarization evolution at the growth interruption for two films of 20 u. c. (interface proximity) and 40 u. c. (no proximity). No polarization evolution at the growth interruption is observed in the thicker layers. In addition, we now emphasize the in-situ monitoring of polarization in the revised title.

3.5) Once these two major issues are addressed, I believe this paper deserves publication.

Answer 3.5:

We have addressed these issues and thank the reviewer once again for helping us improve the manuscript.

3.6) The DFT calculations should be explained a bit more in details in the main text. It is not clear how the polarization gradient is screened in the DFT and if there is a gradient in stoichiometry. Positive charges at the top interface are added to emulate the non-

stoichiometry: this is briefly mentioned in the caption of figure 3 but should be explained in more details in the text.

Answer 3.6:

As suggested, we added the requested details on the DFT calculations in the main text in lines 122-130. The polarization direction imposed in the bottom two u. c. are kept fixed and the polarization in the remaining u. c. is allowed to relax. There is no gradient in stoichiometry.

3.7) Below are additionally minor details that can be addressed:

- Figures 1(a) and 2(b) - I suggest to put the Time as the lower axis and indicate the PTO thickness at the top.

- Also from Figure 1 and 2, the authors say the "the time axis for this growth protocol reveals relaxation of the polarization on the order of 10^2 s" - this is not so clear from the graph (also in supplementary Figure 1)

Answer 3.7:

We followed all these suggestions.

3.8) - Figure 3(a): the scale for the Pb/Ti ratio varies between 1 and 5, and the points vary from 2 to 4. This clearly must be wrong.

Answer 3.8:

XPS evidences an enriched Pb composition throughout the film's thickness with a gradual increase of Pb/Ti ratio towards the outermost part of the samples, in line with the amorphous Pb-rich layer observed by EDX. The intensity of the XPS signal has contributions of the photoelectrons emitted at different depths, thus the Pb/Ti ratio at higher depths can be considered as an average over the entire film thickness. We edited the main text in lines 115-120 to reflect the agreement of the XPS and the EDX analysis.

3.9) - Figure 4(c): this corresponds to the "competitive" configuration where part of the sample has a polarization pointing down and part of the sample has a polarization pointing down. Why then are the OOP values scattered around 0 for PTO, rather than having two clusters with positive and negative values?

Answer 3.9:

The net zero OOP displacement of the multidomain state is the result of averaging of polarization through the sample thickness probed by STEM. We added a comment in the caption of Figure 5.

3.10) - Supplementary section 2: piezoresponse force microscopy: the authors should describe the pattern that was written and mention what are the "virgin" regions. They should also comment on the stability of the written regions. Did they observe some back-switching? Could they also add the corresponding amplitude images? In the figure caption, the authors mixed "b" and "c".

Answer 3.10:

We now include results of the switching retention experiments and the corresponding discussion in Figure 4 and in the main text in lines 139-147 with improved labelling and in-plane response.

3.11)- Supplementary Section 3: the minus signs are missing in the formula for the Schottky barriers: E_c-E_f and E_f-E_v (instead of E_cE_f and E_fE_v).

Answer 3.11:

We corrected this.

3.12)- Supplementary Section: I don't agree when the authors write that the tetragonality according to STEM is uniform throughout both types of samples - it clearly isn't uniform in figures a and b.

Answer 3.12:

We agree. In order to avoid further confusion, we now specifically refer to the tetragonality within the bulk of the PTO layers, excluding the 3 u. c. next to interfaces, which is about 1.06 for both PTO films with competitive and cooperative interfaces.

REVIEWERS' COMMENTS

Reviewer #1 (Remarks to the Author):

I now recommend the publication of this work.

Reviewer #2 (Remarks to the Author):

The authors have taken the time and effort to significantly improve their manuscript in line with the three Reviewers' comments. Specifically, they have added further data to show reproducibility and generality of their observations, additional evidence of the Pb excess at the surface of the films, and ferroelectric switching and retention of the polarization in the two types of films. The clarity of presentation is higher as well. I appreciate the inclusion of further data in the Supplementary information which will help the reader. In my view this paper is now suitable for publication, with just one small addition: In Figure 4, the retention of the polarization is evidenced through PFM images, presumably these are the phase images. As requested by Reviewer 3, it would be useful to show the amplitude images as well for the more complete picture.

Reviewer #3 (Remarks to the Author):

The authors have suitably addressed all my comments. The paper clearly deserves now publication in Nature Communications.

Response to Reviewers: In-situ monitoring of interface proximity effects in ultrathin ferroelectrics (NCOMMS-20-24592)

We would like to thank the Reviewers for their recommendation of publication in Nature Communications and address the remaining concern of Reviewer #2 below.

Reviewer #1 (Remarks to the Author):

I now recommend the publication of this work.

Reviewer #2 (Remarks to the Author):

The authors have taken the time and effort to significantly improve their manuscript in line with the three Reviewers' comments. Specifically, they have added further data to show reproducibility and generality of their observations, additional evidence of the Pb excess at the surface of the films, and ferroelectric switching and retention of the polarization in the two types of films. The clarity of presentation is higher as well. I appreciate the inclusion of further data in the Supplementary information which will help the reader. In my view this paper is now suitable for publication, with just one small addition: In Figure 4, the retention of the polarization is evidenced through PFM images, presumably these are the phase images. As requested by Reviewer 3, it would be useful to show the amplitude images as well for the more complete picture.

In the Figure 4, we show the PFM channels as in-phase signal for out-of-plane (main panel). The in-plane (inset) information comes from the quadrature signal. This indeed gives the full picture of the retention of the polarization in our films. We now specify the PFM channels in the caption of Figure 4.

Reviewer #3 (Remarks to the Author):

The authors have suitably addressed all my comments. The paper clearly deserves now publication in Nature Communications.